# Analysis of Energy Flow in a Mid-Sized Electric Passenger Vehicle in Urban Driving Conditions

Youngkuk An [1,†], Byeonggyu Yang [1,†], Jinil Park [1,*], Jonghwa Lee [1] and Kyoungseok Park [2,*]

[1] Department of Mechanical Engineering, Ajou University, Suwon 16499, Republic of Korea; stingyk@ajou.ac.kr (Y.A.); ybk95@ajou.ac.kr (B.Y.); jlee@ajou.ac.kr (J.L.)

[2] Department of Mechanical System Engineering, Kumoh National Institute of Technology, Gumi 39177, Republic of Korea

[*] Correspondence: jpark@ajou.ac.kr (J.P.); kspark@kumoh.ac.kr (K.P.); Tel.: +82-31-219-2337 (J.P.); +82-54-478-7322 (K.P.)

[†] These authors contributed equally to this work.

**Abstract:** Because of emissions of exhaust gases, global warming is proceeding, and air pollution has increased. Thus, many countries are manufacturing eco-friendly vehicles, including electric vehicles. However, the range of electric vehicles is less than the range of internal combustion engine vehicles, so electric vehicle production is being disrupted. Thus, it is necessary to analyze the energy flow of electric vehicles. Therefore, to analyze energy flow of electric vehicles, this study suggested an energy flow structure first, then modeled the energy flow of the vehicle, dividing it into battery, inverter and motor, reduction gear and differential, and wheel parts. This study selected a test vehicle, drove in urban driving conditions and measured data. Then, this study calculated energy flow using MATLAB/SIMULINK in real time, and calculated and analyzed energy loss of each of the vehicle's parts using the calculated data.

**Keywords:** energy flow; EV; modeling; UDDS; energy loss; energy flow analysis

## 1. Introduction

In response to global warming and deteriorating air quality, policies to control vehicle emissions are being implemented worldwide. Accordingly, each country is expanding the supply of eco-friendly vehicles, excluding hybrid and plug-in hybrid vehicles using internal combustion engines in the long term, and expanding the supply of electric vehicles and hydrogen fuel cell vehicles that do not emit emissions [1,2].

However, the spread of electric vehicles is hindered by various factors, and one of them is the short driving range compared to internal combustion engines [3]. Short mileage leads to frequent charging, and long charging times and poor charging infrastructure increase the time spent on charging, which causes people to avoid electric vehicles. Therefore, it is necessary to study how to increase the mileage by reducing the energy loss of these vehicles.

In the past, studies on electric vehicles have focused primarily on the impact of traffic conditions and vehicle driving conditions on the vehicle's overall energy consumption [4,5]. In another study, analysis of the energy efficiency of electric vehicles was conducted according to the season and driver's propensity [6,7]. Studies on energy consumption of electric vehicles include a study on the difference in energy consumption according to the ambient temperature [8], as well as studies on energy consumption models by vehicle speed, acceleration functions and driving modes [9,10].

On the other hand, research trends in South Korea suggest a shorter test method than the current energy efficiency test method for electric vehicles, and verify its feasibility [11]. Others study the charging mileage according to the outside temperature and the mileage according to the charging speed [12]. However, most of the studies only analyze the overall

energy efficiency according to the change of factors, and the energy flow analysis for each detailed factor is insufficient. Therefore, the purpose of this study is to collect the data necessary for energy flow analysis, analyze the data, and study the energy flow for each detailed element. To this end, it is necessary to model the energy flow, calculate it, and analyze the loss items.

In this study, the energy flow of an electric vehicle was modeled. The energy flow was calculated through experimental data obtained by driving in a city environment, and the driving and braking energy of vehicle and energy loss were analyzed.

To this end, the specifications of the test vehicle were first marked, and the energy flow of the vehicle was modeled. As for the energy flow of the test vehicle, when the vehicle is accelerating and cruising, the electric power energy from the battery is converted into mechanical energy through the inverter and motor, and then transmitted to the wheel through the reducer and differential. During deceleration, the inertial energy of the vehicle is converted into electrical energy through the wheels, reducer and differential, motor and inverter, and then stored in the battery.

For energy flow analysis, test data were derived through the urban driving mode. Afterward, the energy flow was modeled, and based on this, the energy flow in each item, such as the battery and motor, were calculated using MATLAB/SIMULINK. In addition, the energy loss in each item was analyzed.

## 2. Test Vehicle

The test vehicle is the 2021 'Model Y', an electric car from Tesla Motors, and the detailed trim is 'Long Range'.

The specifications of the test vehicle are as follows in Table 1.

**Table 1.** Test vehicle specification [13].

| List | Specification |
| --- | --- |
| Curb weight [kg] | 2000 |
| Front motor max power [kW] | 158 |
| Rear motor max power [kW] | 208 |
| Front motor max torque [Nm] | 240 |
| Rear motor max torque [Nm] | 353 |
| Battery capacity [kWh] | 75 |
| Battery type | Lithium ion |
| Tire specification | 255/45/R19 |
| Length [mm] | 4750 |
| Width [mm] | 1920 |
| Height [mm] | 1625 |
| Range [km] | 511 |

The test vehicle is four-wheel drive, and inverters and motors at the front and rear of the vehicle drive the front and rear wheels, respectively.

## 3. Energy Flow Modeling

The energy flow of the vehicle is as follows. During vehicle acceleration and cruising, electric power energy from the battery is transferred to the inverter and converted into mechanical energy through the motor. Then, the mechanical energy of the motor is transmitted to the wheels of the vehicle through the reducer and differential, and the energy remaining after being consumed as driving resistance is stored as inertial energy of the vehicle.

During deceleration, the energy remaining after excluding the energy consumed by friction braking and driving resistance from the inertial energy of the vehicle is transmitted to the motor through the reducer and differential. Afterwards, the motor acts as a generator to convert mechanical energy into electrical energy, which is then stored in a battery through an inverter.

In this study, we classified the energy flow into detailed consumption components such as the battery, inverter and motor, reducer and differential, and wheels, and formulated equations for the energy transfer in each part. We used these equations to analyze the energy flow of the entire electric vehicle.

The nomenclature of the variables used in the equations is shown in Table 2.

**Table 2.** Nomenclature.

| Symbol | Definition | Unit |
|---|---|---|
| $E_{B,out}$ | Battery discharge energy in acceleration and cruise | J |
| $E_{B,Loss}$ | Energy lost due to battery internal resistance in acceleration and cruise | J |
| $E_{B,in,Regen}$ | Battery charge energy in deceleration | J |
| $E_{B,Loss,Regen}$ | Energy lost due to battery internal resistance in deceleration | J |
| $E_{DC-DC}$ | Energy supplied to DC–DC converter in acceleration and cruise | J |
| $E_{I,in}$ | Energy supplied to Inverter in acceleration and cruise | J |
| $E_{I,out,Regen}$ | Output energy from inverter in deceleration | J |
| $E_{M,out}$ | Output energy from motor in acceleration and cruise | J |
| $E_{M-I,Loss}$ | Loss energy from Inverter and motor in acceleration and cruise | J |
| $E_{M,in,Regen}$ | Energy supplied to Motor in deceleration | J |
| $E_{M-I,Loss,Regen}$ | Loss energy from Inverter and motor in deceleration | J |
| $E_{D,in}$ | Energy supplied to reducer/differential gear in acceleration and cruise | J |
| $E_{D,in,Regen}$ | Energy supplied to reducer/differential gear in deceleration | J |
| $E_{D,out}$ | Output energy from reducer/differential gear in acceleration and cruise | J |
| $E_{gear,Loss}$ | Loss energy due to reducer/differential gear friction in acceleration and cruise | J |
| $E_{D,out,Regen}$ | Output energy from reducer/differential gear in deceleration | J |
| $E_{gear,Loss,Regen}$ | Loss energy due to reducer/differential gear friction in deceleration and cruise | J |
| $E_{W,I}$ | Loss energy due to Axle and wheel inertia in acceleration and cruise | J |
| $E_{RL}$ | Loss energy due to Road load in acceleration and cruise | J |
| $E_{RL,Regen}$ | Loss energy due to Road load in deceleration | J |
| $E_{VI}$ | Energy of vehicle inertia | J |
| $E_{Brake\ pad}$ | | J |
| $I_B$ | Battery current | A |
| $I_{DC-DC}$ | DC–DC converter current in acceleration and cruise | A |
| $I_{DC-DC,Regen}$ | DC–DC converter current in deceleration | A |
| $I_I$ | Inverter current | A |
| $I_{Tire}$ | Tire rotational inertia | W |
| $I_{axle}$ | Driveshaft rotational inertia | W |
| $P_B$ | Battery electric power in acceleration and cruise | W |
| $P_{B,Loss}$ | Battery loss power in acceleration and cruise | W |
| $P_{B,Loss,Regen}$ | Battery loss power in deceleration | W |
| $P_{brake,pad}$ | Friction brake loss power | W |
| $P_{B,Regen}$ | Battery electric power in deceleration | W |
| $P_{DC-DC}$ | DC–DC converter power in acceleration and cruise | W |
| $P_{DC-DC,Regen}$ | DC–DC converter power in deceleration | W |
| $P_{I,in}$ | Inverter input power in acceleration and cruise | W |
| $P_{I,out,Regen}$ | Inverter output power in deceleration | W |
| $P_{M,in}$ | Motor input power in acceleration and cruise | W |
| $P_{M,out}$ | Motor output power in acceleration and cruise | W |
| $P_{M,in,Regen}$ | Motor input power in deceleration | W |
| $P_{M,out,Regen}$ | Motor output power in deceleration | W |
| $P_{D,in}$ | Reduction gear input power in acceleration and cruise | W |
| $P_{D,out,Regen}$ | Reduction gear output power in deceleration | W |
| $P_{D,out}$ | Differential output power in acceleration and cruise | W |
| $P_{D,in,Regen}$ | Differential input power in deceleration | W |
| $P_{VI}$ | Vehicle inertia power | W |
| $P_{RL}$ | Loss power due to road load in acceleration and cruise | W |
| $P_{RL,Regen}$ | Loss power due to road load in deceleration | W |

**Table 2.** *Cont.*

| Symbol | Definition | Unit |
|---|---|---|
| $P_{W,I}$ | Loss power due to axle and wheel rotational inertia | W |
| $R_{int}$ | Battery internal resistance | $\Omega$ |
| $V_I$ | Inverter voltage | V |
| $\omega_M$ | Motor revolution speed | Rpm |
| $T_D$ | Driveshaft torque | Nm |
| $T_M$ | Motor torque | Nm |
| $\omega_D$ | Driveshaft revolution speed | Rpm |
| $M_V$ | Vehicle mass | Kg |
| $V$ | Vehicle speed | m/s |
| $a$ | Vehicle acceleration | $m/s^2$ |
| $F_0$ | Road load coefficient | N |
| $F_1$ | Road load coefficient | N/kph |
| $F_2$ | Road load coefficient | $N/kph^2$ |
| $V_k$ | Vehicle speed | kph |

*3.1. Energy Flow Modeling*

A schematic diagram of the overall energy flow modeling of the vehicle is shown as Figure 1.

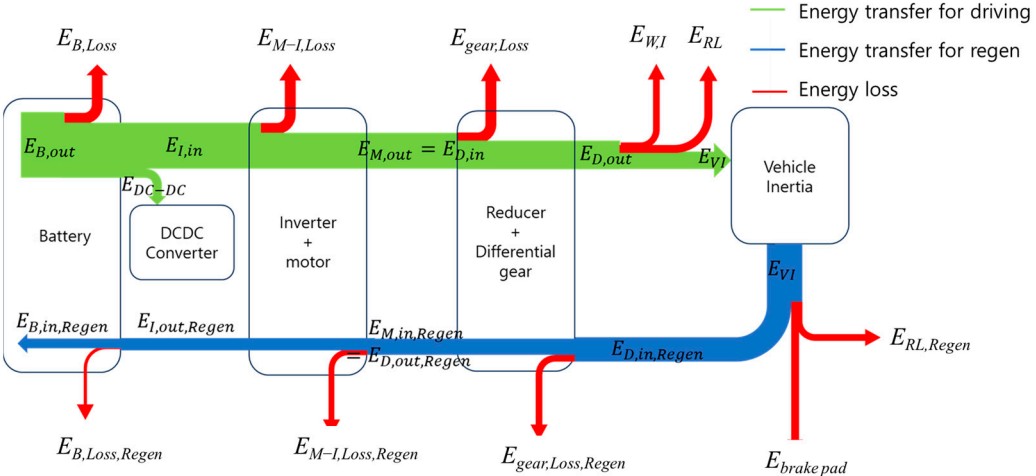

**Figure 1.** Vehicle total energy flow.

The green arrow represents the energy flow during vehicle acceleration and cruising, and the red arrow represents the energy flow during vehicle deceleration. The yellow arrow indicates the flow of inertial energy of the motor, reducer, and differential; inertial energy is stored in the rotor, reducer, and gear of the differential during acceleration and cruising of the vehicle, and then returned when the vehicle is driven at a reduced speed. At this time, the energy supplied to the DC–DC converter of the vehicle is used to drive the vehicle's electrical accessories, but the energy is very small, so loss efficiency is not considered [14].

3.1.1. Battery

The energy flow of the battery was modeled considering the internal resistance of the battery and DC–DC converter. The electric power energy of the battery is as follows:

$$P_B = P_{B,\,Loss} + P_{DC-DC} + P_{I,\,in}\ (\text{Acceleration/Cruising}) \tag{1}$$

$$P_{B,Regen} = P_{I,Out,Regen} + P_{B,Loss,Regen}\ (\text{Deceleration}) \tag{2}$$

The energy loss power due to Battery internal resistance is as follows:

$$P_{B,Loss} = \begin{cases} I_B{}^2 \cdot R_{int} & (I_B \geq 0) \\ 0 & (I_B < 0) \end{cases} \text{(Acceleration/Cruising)} \tag{3}$$

$$P_{B,Loss,Regen} = \begin{cases} 0 & (I_B \geq 0) \\ I_B{}^2 \cdot R_{int} & (I_B < 0) \end{cases} \text{(Deceleration)} \tag{4}$$

The electric power energy consumed by the DC–DC converter is as follows:

$$P_{DC-DC} = I_{DC-DC} + V_{DC-DC} \text{ (Acceleration/Cruising)} \tag{5}$$

$$P_{DC-DC,Regen} = 0 \text{ (Deceleration)} \tag{6}$$

When the vehicle air conditioner is not operated, the energy regenerated during vehicle braking does not flow to the DC–DC converter. In this study, the air conditioner was not operated during the vehicle test. In the case of the BMW i3, 26% of the energy consumed from the battery when operating the air conditioner is used to operate the electric heater [15]. Therefore, in this study, when the air conditioner is operated, it is expected that the loss of the air conditioner will occupy a large portion of the loss items.

The inverter input power during vehicle acceleration/cruising and the inverter output power during deceleration driving are as follows:

$$P_{I,in} = \begin{cases} I_I \cdot V_I & (I_I \geq 0) \\ 0 & (I_I < 0) \end{cases} \text{(Acceleration/Cruising)} \tag{7}$$

$$P_{I,out,Regen} = \begin{cases} 0 & (I_I \geq 0) \\ I_I \cdot V_I & (I_I < 0) \end{cases} \text{(Deceleration)} \tag{8}$$

### 3.1.2. Inverter and Motor

Since there is no current and voltage data between the inverter and the motor, the inverter and the motor are configured as an integrated model. During vehicle acceleration and cruising, the difference between the input energy of the inverter and the output energy of the motor was regarded as the total loss energy of the inverter and motor. During vehicle deceleration, the difference between the input energy of the motor and the output energy of the inverter was regarded as the total loss energy of the inverter and motor.

The output power of the motor during vehicle acceleration and cruising and the input power of the motor during deceleration are as follows:

$$P_{M,out} = \begin{cases} T_M \cdot \omega_M & (T_M \geq 0) \\ 0 & (T_M < 0) \end{cases} \text{(Acceleration/Cruising)} \tag{9}$$

$$P_{M,in,Regen} = \begin{cases} 0 & (T_M \geq 0) \\ T_M \cdot \omega_M & (T_M < 0) \end{cases} \text{(Deceleration)} \tag{10}$$

Using the equation defined above, Equations (7)–(10), the input and output power of the inverter/motor can be defined as follows:

$$P_{M,out} = P_I \eta_{inverter/motor} \text{ (Acceleration/Cruising)} \tag{11}$$

$$P_{I,out,Regen} = P_{M,in,Regen} \eta_{inverter/motor} \text{ (Deceleration)} \tag{12}$$

$$\eta_{inverter/motor} = \begin{cases} \dfrac{P_{M,out}}{P_I} = \dfrac{T_M \cdot \omega_M}{V_I I_I} & \text{(Acceleration/Cruising)} \\ \dfrac{P_{I,out,Regen}}{P_{M,in,Regen}} = \dfrac{V_I \cdot I_I}{T_M \omega_M} & \text{(Deceleration)} \end{cases} \tag{13}$$

### 3.1.3. Reducer and Differential

The reducer and differential are configured as an integrated model. During vehicle acceleration and cruising, the difference between the input energy of the reducer and the output energy of the differential was regarded as the total energy loss. During vehicle deceleration, the difference between the input energy of the differential and the output energy of the reducer is calculated as the total energy loss. During vehicle acceleration and cruising, the input energy of the reduction gear is the same as the output energy of the reducer, which is equal to the input energy of the motor. The output power of the differential during vehicle acceleration and cruising and the input power of the differential during deceleration are as follows:

$$P_{D,out} = \begin{cases} T_D \cdot \omega_D & (T_D \geq 0) \\ 0 & (T_D < 0) \end{cases} \text{(Acceleration/Cruising)} \tag{14}$$

$$P_{D,in,Regen} = \begin{cases} 0 & (T_D \geq 0) \\ T_D \cdot \omega_D & (T_D < 0) \end{cases} \text{(Deceleration)} \tag{15}$$

The relational expression for the input and output power of the reducer and differential gear while the vehicle is running can be defined as follows by Equations (9), (10), (14) and (15).

$$P_{D,out} = P_{M,out} \eta_{reducer,diff} \text{ (Acceleration/Cruising)} \tag{16}$$

$$P_{M,in,Regen} = P_{D,in,Regen} \eta_{reducer,diff} \text{ (Deceleration)} \tag{17}$$

$$\eta_{reducer,diff} = \begin{cases} \frac{P_{D,out}}{P_{M,out}} = \frac{T_D \cdot \omega_D}{T_M \cdot \omega_M} & \text{(Acceleration/Cruising)} \\ \frac{P_{M,in,Regen}}{P_{D,in,Regen}} = \frac{T_M \cdot \omega_M}{T_D \cdot \omega_D} & \text{(Deceleration)} \end{cases} \tag{18}$$

The output energy of the differential during vehicle acceleration and cruising is the power energy transmitted to the wheels, and the input energy of the differential during deceleration is the regenerated energy from the wheels. This can be expressed as follows:

$$P_{D,out} = P_{VI} + P_{W,I} + P_{RL} \text{ (Acceleration/Cruising)} \tag{19}$$

$$P_{D,in,Regen} = P_{VI} + P_{W,I} - P_{RL,Regen} - P_{brake,pad} \text{ (Deceleration)} \tag{20}$$

### 3.1.4. Wheel

For the wheel, the rotational inertial energy of the was assumed. This value, the driving resistance of the vehicle, vehicle inertia, and the rotational inertial energy of the wheel were reflected in the model energy flow in the wheel.

Vehicle inertia and rotational inertial energy are as follows:

$$P_{VI} = M_V \cdot V \cdot a \tag{21}$$

$$P_{W,I} = I_{axle} + I_{Tire} \tag{22}$$

The rotational inertial energy of the driveshaft was calculated by assuming the material, radius, and length of the driveshaft. It is assumed that the material of the driveshaft is steel, the diameter is 100 mm, and the length is the value obtained by subtracting the tire width 255 mm from the wheel track of the vehicle.

The driving resistance and the frictional braking energy loss during acceleration, cruising, and deceleration are as follows:

$$P_{RL} = \begin{cases} \left(F_0 + F_1 V_k + F_2 V_k{}^2\right)V & (P_{VI} + P_{W,I} + P_{RL} \geq 0) \\ 0 & (P_{VI} + P_{W,I} + P_{RL} < 0) \end{cases} \text{(Acceleration/Cruising)} \tag{23}$$

$$P_{RL} = \begin{cases} 0 & (P_{VI} + P_{W,I} + P_{RL} \geq 0) \\ \left(F_0 + F_1 V_k + F_2 V_k{}^2\right) V & (P_{VI} + P_{W,I} + P_{RL} < 0) \end{cases} \text{(Deceleration)} \quad (24)$$

$$P_{brake,pad} = P_{VI} + P_{W,I} - P_{RL,Regen} - P_{D,in,Rengen} \quad (25)$$

The frictional braking loss energy was calculated after all other factors were calculated.

## 4. Analysis of Energy Flow

After driving in the test-driving mode, the data were obtained, and the energy flow in each item was calculated using MALAB. MATLAB was used for interpolating time-series data and calculating the energy of each component based on the equations in Section 3. Additionally, it was utilized to apply a low-pass filter to the raw data from the torque sensor and to use a Gaussian filter on the vehicle speed for calculating acceleration. The reason for using MATLAB even for simple calculations is to facilitate data integration when building a vehicle simulation model in the future using SIMULINK for further research.

### 4.1. Test Drive Mode

The test-driving mode is a mode in which the UDDS driving mode is repeatedly driven twice. The UDDS driving mode, also called FTP-72 mode, is a fuel economy test mode that implements urban driving speed patterns. The speed pattern of the UDDS mode is shown as Figure 2.

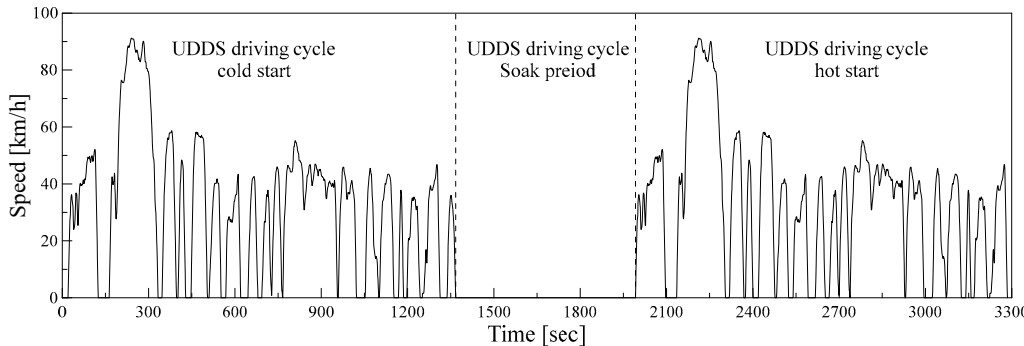

**Figure 2.** Vehicle speed profile of test-driving cycle.

The first run is cold start, and the next run is hot start. During the test run, the outside temperature is 25 degrees Celsius, and the test run time is about 3300 s. The air conditioner was not operated.

### 4.2. Test Drive Data

The schematic diagram for the operation and energy flow of an electric vehicle is as shown in Figure 3. Table 3 shows the list of collected data for energy flow analysis of electric vehicles and related collection equipment.

As seen in Table 3, most of the data can be collected from the CAN logger. However, the current of the front motor, rear motor, and air compressor in the high-voltage system cannot be found in the CAN list. Therefore, current sensors were installed to measure their currents, and to measure the axle torque, strain gauges were installed on the driveshaft, and torque telemetry was used to measure the torque.

The data for the test drive were collected using the CAN information of the vehicle, current sensors, and torque telemetry.

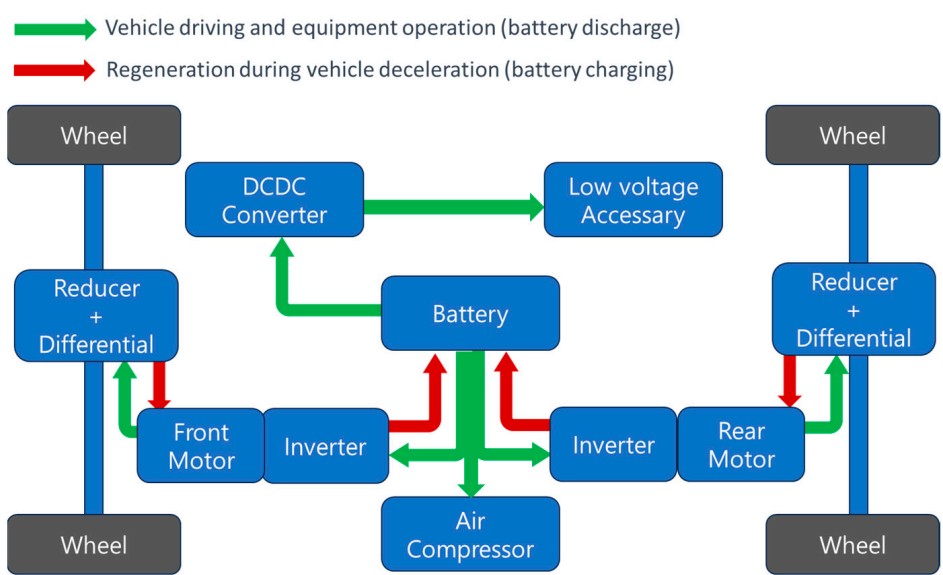

**Figure 3.** Energy flow schematic of electric vehicle.

**Table 3.** Collected data list.

| List | Equipment |
|---|---|
| Battery voltage [V] | CAN logger |
| Battery current [A] | CAN logger |
| DC–DC converter input current [A] | CAN logger |
| DC–DC converter output current [A] | CAN logger |
| Low voltage battery voltage [V] | CAN logger |
| Compressor current [A] | Current sensor |
| Front inverter/motor current [A] | Current sensor |
| Rear inverter/motor current [A] | Current sensor |
| Front motor torque [Nm] | CAN logger |
| Front motor speed [rpm] | CAN logger |
| Rear motor torque [Nm] | CAN logger |
| Rear motor speed [rpm] | CAN logger |
| Front axle torque left/right [Nm] | Torque Sensor |
| Rear axle torque left/right [Nm] | Torque Sensor |
| Front axle speed [rpm] | CAN logger |
| Rear axle speed [rpm] | CAN logger |
| Vehicle speed [km/h] | CAN logger |

### 4.2.1. CAN Data

The collection of CAN data was performed using the ETAS ES-582.2 device.

### 4.2.2. Current Sensor

To measure the current of the front and rear motors and the air compressor, the SCV-U2 current sensor was used. However, in this study, the air conditioner was not operated during the UDDS mode driving, so the data from the air compressor current sensor was not used. The validation of the current sensor was confirmed by checking whether it satisfied the following equation:

$$I_B = I_I + I_{DCDC} \tag{26}$$

To validate Equation (22), the graph comparing battery current, motor current for front and rear shaft, and DC–DC converter current data from the UDDS mode driving is shown Figure 4.

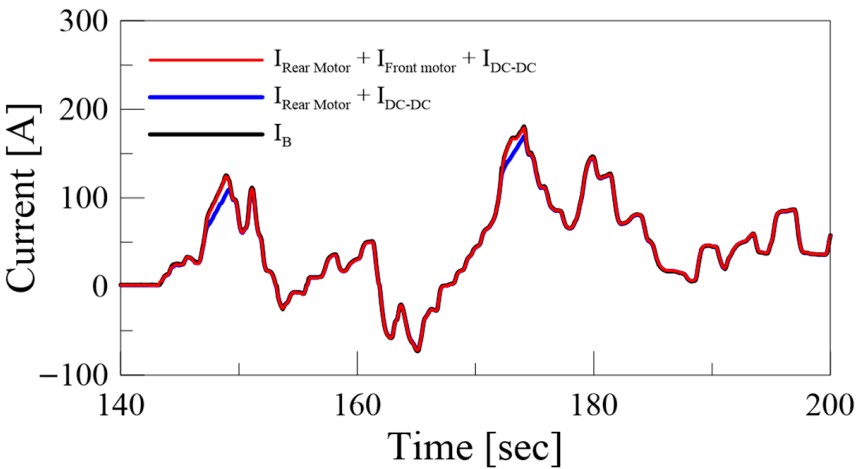

**Figure 4.** Validation of current sensor.

The error between the sum of motor current sensor values and the DC–DC converter current value in the entire UDDS mode is approximately 0.2%. Therefore, it can be concluded that there are no issues with the energy flow analysis for each motor current sensor value.

### 4.2.3. Torque Sensor

The electrical energy supplied to the motor is converted into mechanical energy within the motor. The converted mechanical energy is then transmitted through the differential gear to the driveshaft. To measure the mechanical energy, strain gauges were attached to the driveshaft, and torque was measured using torque telemetry (Axon J1).

To apply the torque sensor, a 100 kg weight was attached to a one meter arm, and the sensor was calibrated by applying a predetermined torque. The results are as follows in Figure 5.

### 4.2.4. Data Acquisition

The data collected to analyze the energy flow of mechanical power transmission for the electric vehicle are shown in Figure 6, while the data collected to analyze the energy flow of the electrical system for the electric vehicle are presented in Figure 7.

### *4.3. Energy Flow in Test Drive Mode*

Using the model equations of each part of an electric vehicle described in Section 3 and the acquired data in Section 4.2, the energy transfer amount of each part of the electric vehicle energy flow diagram introduced in Figure 1 can be calculated.

### 4.3.1. Battery

Battery energy transfer is divided into input/output energy from the battery and energy loss due to internal resistance.

The internal resistance of a battery can be obtained from a relation expression of battery temperature, battery current, and battery charge amount [16]. The energy loss due to the internal resistance of the battery can be obtained by Equations (2) and (3).

The output energy of the battery is the amount of energy discharged from the battery to drive the vehicle during acceleration and cruising, and can be calculated by Equation (1).

The input energy of the battery is the energy transferred to the battery when the vehicle decelerates and can be calculated by the Equation (2).

The calculated real-time energy transfer amounts related to the battery are shown in Figure 8, and the total amount is shown in Table 4.

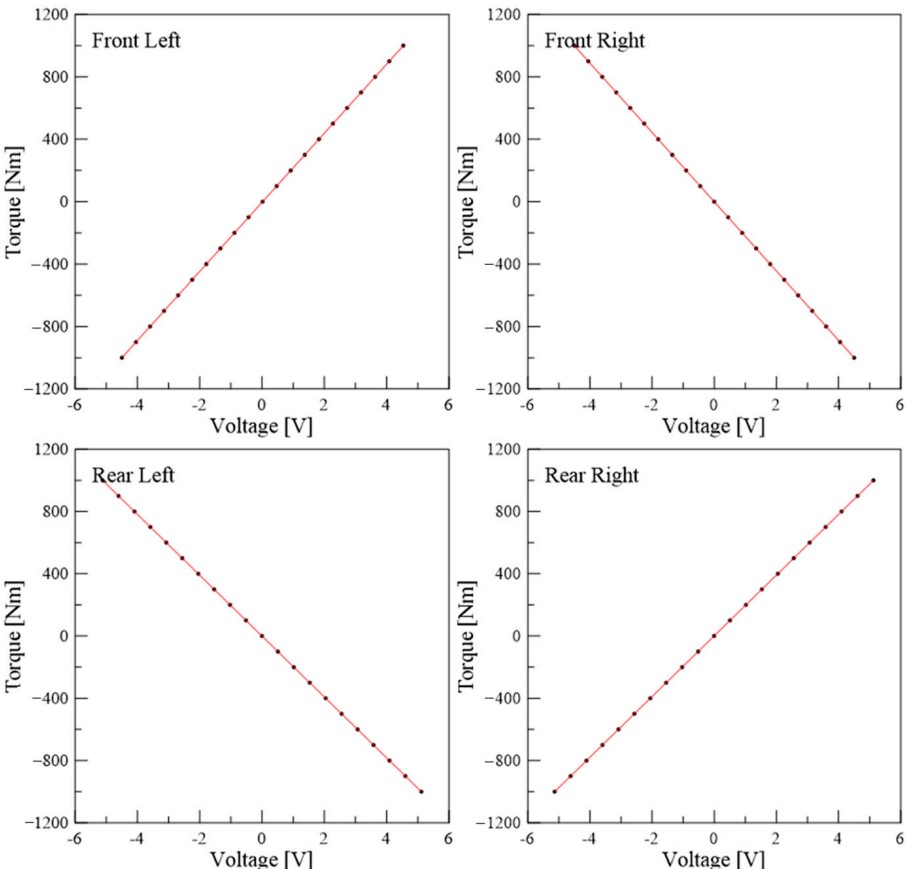

**Figure 5.** Driveshaft torque calibration data.

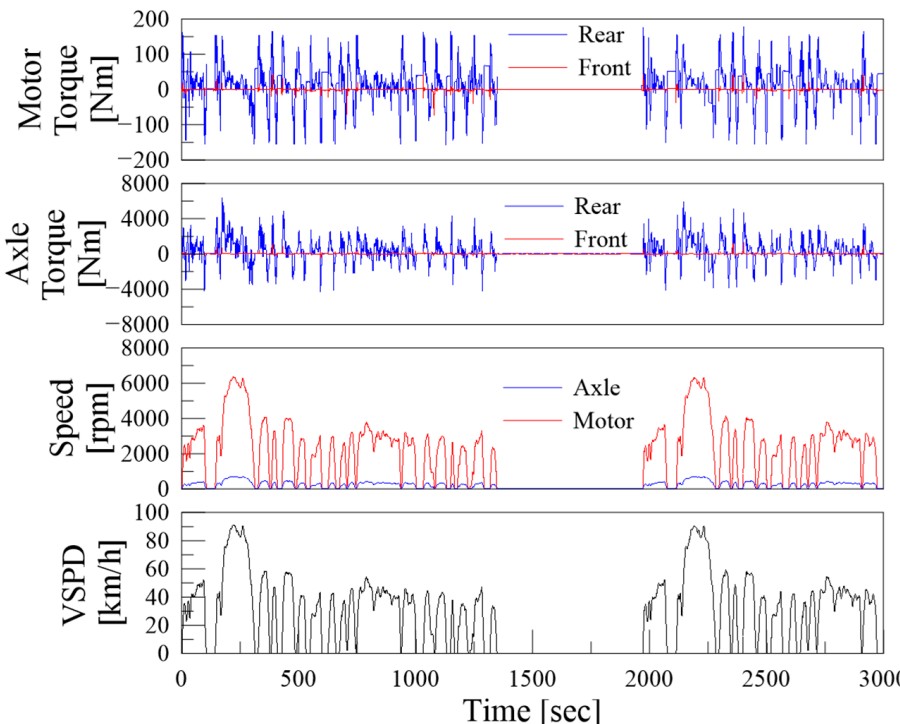

**Figure 6.** Data acquisition of energy flow of mechanical transmission.

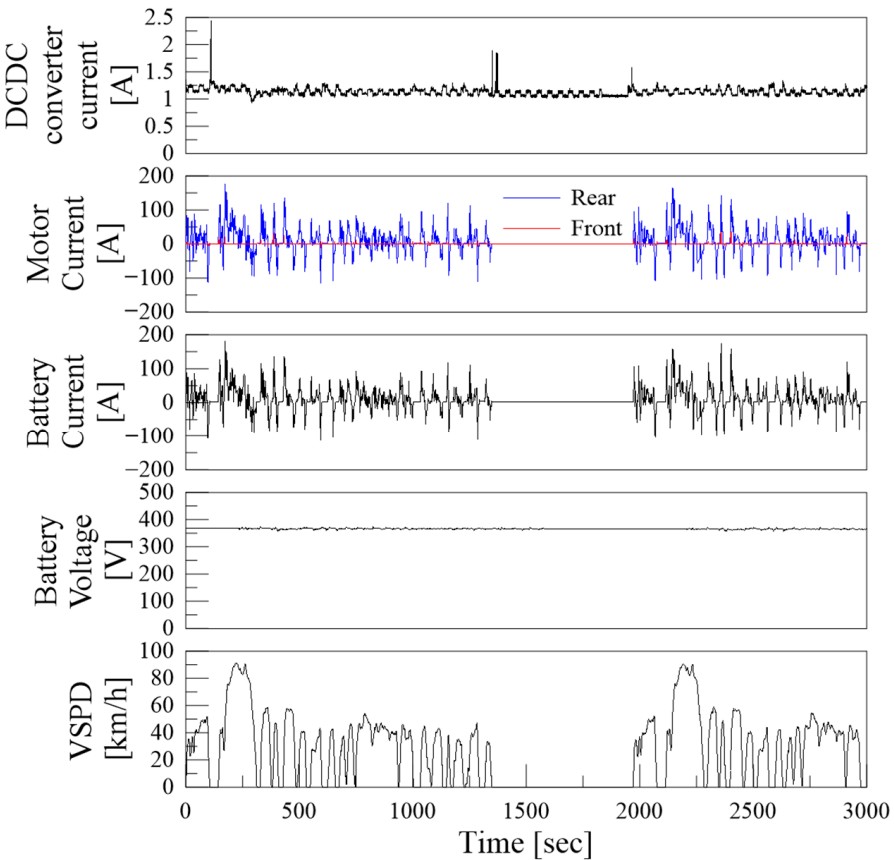

**Figure 7.** Data acquisition of energy flow of the electrical system.

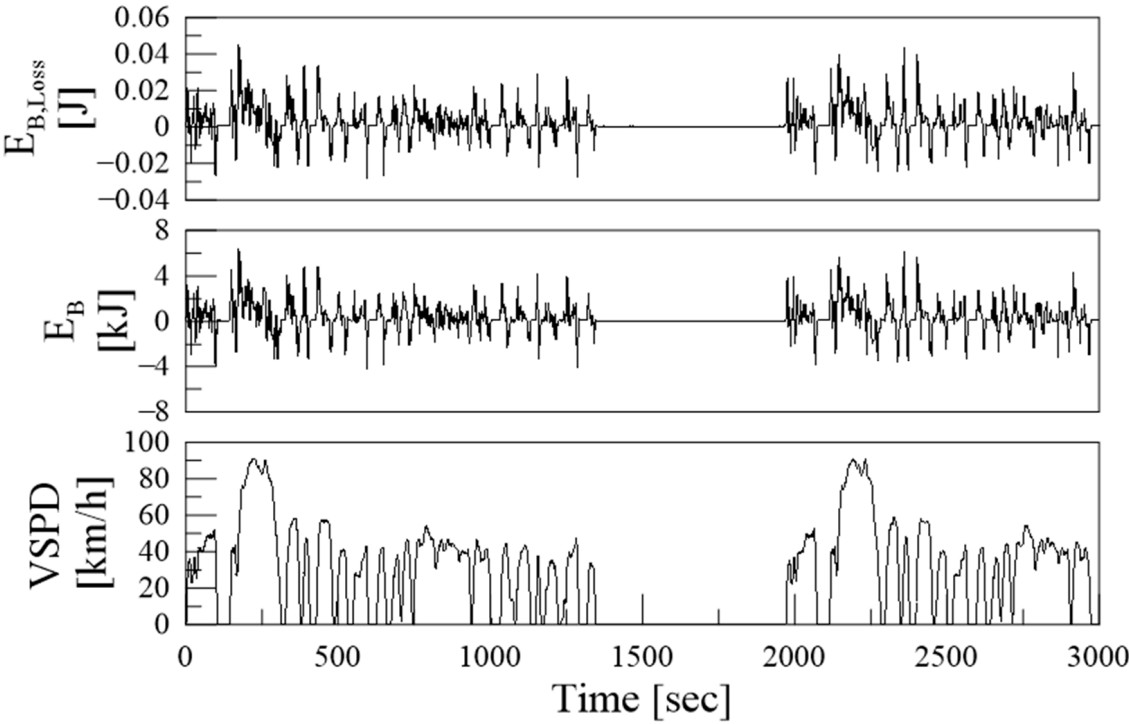

**Figure 8.** Energy transfer amounts related to the battery.

**Table 4.** Total energy amount related to the battery.

| List | Energy [MJ] |
|---|---|
| $E_{B,out}$ | 18.17 |
| $E_{B,in,Regen}$ | 5.99 |
| $E_{B,Loss}$ | 0.24 |
| $E_{B,Loss,Regen}$ | 0.08 |

### 4.3.2. Electrical System

During acceleration and cruise driving, discharged energy from the battery is supplied to the DC–DC converter and the inverter that controls the front/rear motors. The energy delivered to the DC–DC converter can be obtained by Equation (5).

During acceleration and cruising, the inverter supplies current to the motor, and the motor converts the supplied electric energy into mechanical energy and transmits it to the reduction gear. However, when the vehicle is decelerating, the motor receives mechanical energy from the reducer and converts it into electrical energy, and the inverter transfers the supplied electrical energy to the battery to perform regeneration.

In this study, energy transfer between the inverter and the motor was not considered due to the difficulty of data measurement, and the data analysis was conducted by assuming the inverter and the motor as one device.

The input energy of the motor/inverter during vehicle acceleration and cruising is calculated by Equation (7), and the output energy is calculated by Equation (9). The energy loss of the motor/inverter is as follows:

$$E_{M-I,Loss} = E_{I,in} - E_{M,out} \text{ (Acceleration/Cruising)} \tag{27}$$

$$E_{M-I,Loss,Regen} = E_{M,in,Regen} - E_{I,in,Regen} \text{ (Deceleration)} \tag{28}$$

The calculated real-time energy transfer of the electrical system is shown in Figure 9, and the total energy is shown in Table 5.

**Table 5.** Total energy amount of electrical system.

| List | Energy [MJ] |
|---|---|
| $E_{I,in}$ | 16.81 |
| $E_{I,out,Regen}$ | 6.04 |
| $E_{M,out}$ | 13.82 |
| $E_{M,in,Regen}$ | 6.58 |
| $E_{DC-DC}$ | 1.13 |
| $E_{M-I,Loss}$ | 2.99 |
| $E_{M-I,Loss,Regen}$ | 0.50 |

At this time, the efficiency of the motor/inverter can be calculated using Equation (13), and the calculated value is shown in Figure 10.

### 4.3.3. Reducer and Differential Gear

The reducer and differential gear transfer the mechanical energy received from the motor to the axle shaft during vehicle acceleration and cruising. At this time, the input energy of the reducer/differential gear is equal to the output energy of the motor. The output energy is calculated by Equation (14).

During deceleration, the energy received from the axle shaft is transferred to the motor, the input energy of the reducer/differential gear is calculated by Equation (15), and the output energy is equal to the input energy of the motor/inverter.

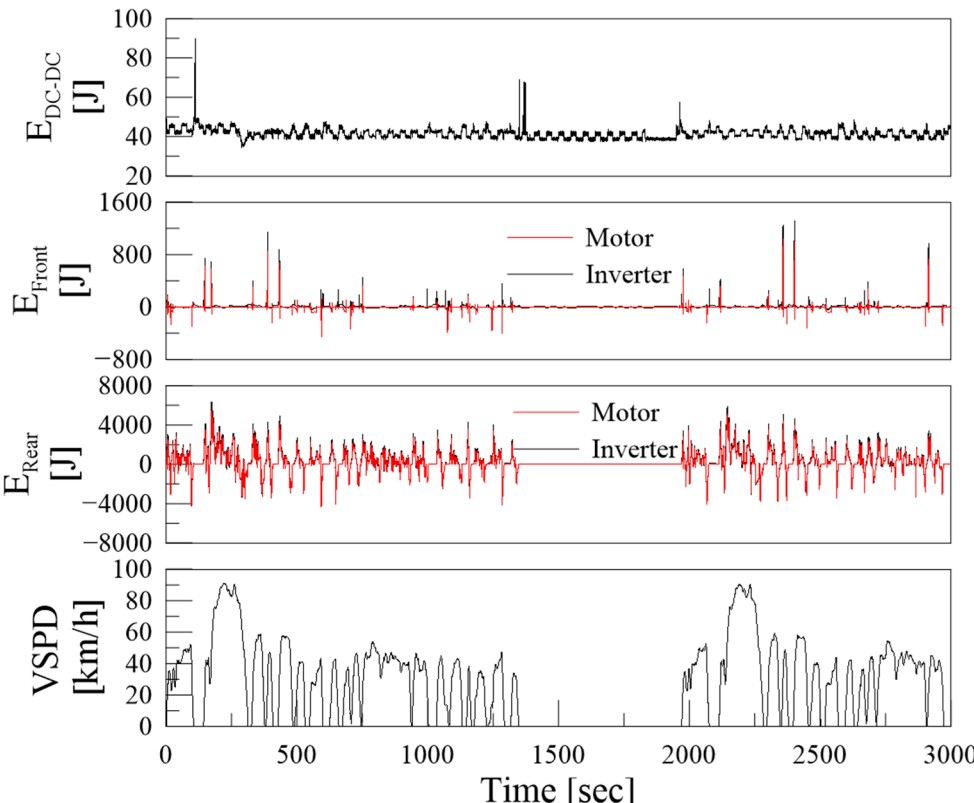

**Figure 9.** Energy transfer amounts of electrical system.

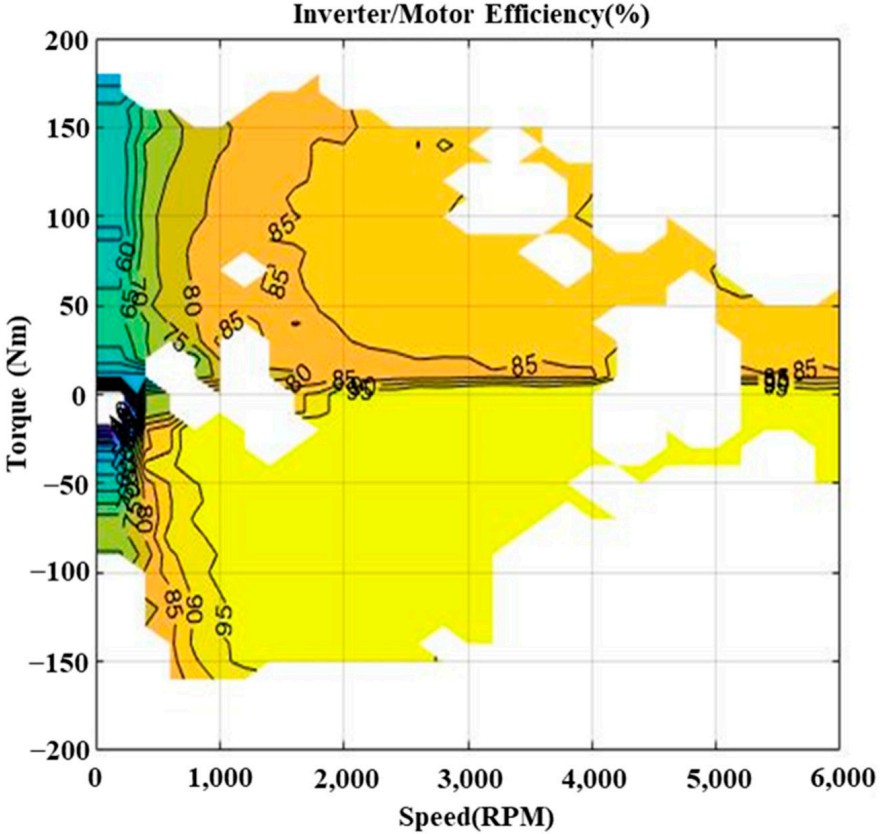

**Figure 10.** Efficiency of the inverter/motor.

The energy loss due to the reducer/differential gear is calculated as follows:

$$E_{gear,Loss} = E_{M,out} - E_{D,out} (Acceleration/Cruising) \tag{29}$$

$$E_{gear,Loss,Regen} = E_{D,in,Regen} - E_{Motor,in,Regen} (Deceleration) \tag{30}$$

The calculated input/output energy of the real-time reducer/differential gear is shown in Figure 11. The total energy of reducer/differential gear is shown in Table 6.

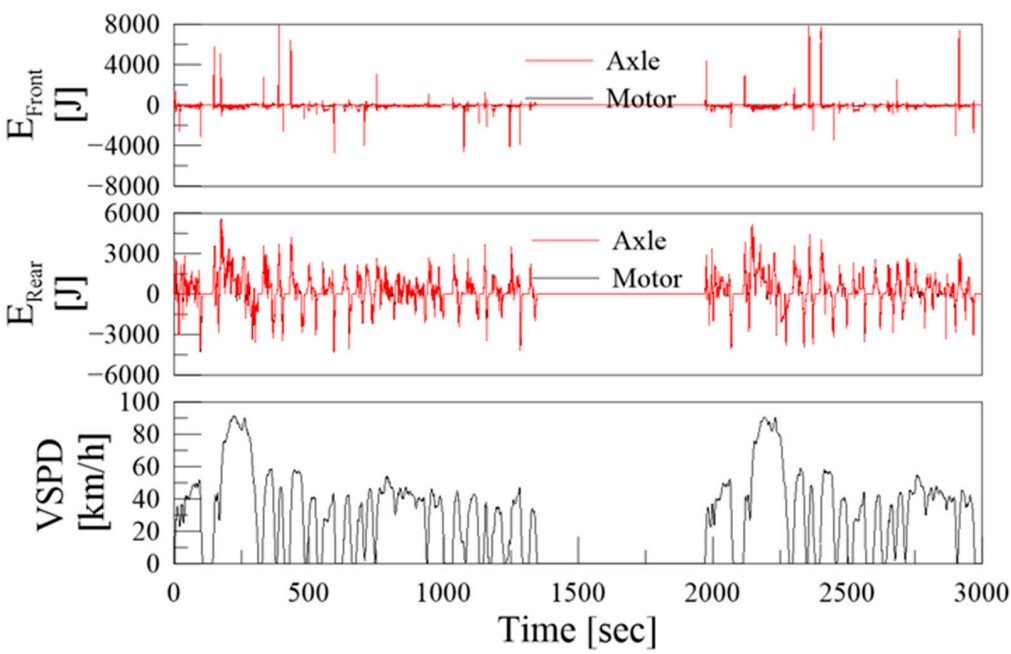

**Figure 11.** Energy transfer amounts of reducer/differential gear.

**Table 6.** Total energy amount of reducer/differential gear.

| List | Energy [MJ] |
|---|---|
| $E_{D,in} = E_{M,out}$ | 13.82 |
| $E_{D,out}$ | 13.59 |
| $E_{gear,Loss}$ | 0.22 |
| $E_{D,in,Regen}$ | 7.09 |
| $E_{D,out,Regen} = E_{M,in,Regen}$ | 6.58 |
| $E_{gear,Loss,Regen}$ | 0.51 |

### 4.3.4. Road Load, Vehicle Inertia, and Other Losses

The energy transmitted from the driveshaft axle during vehicle acceleration and cruising is divided into the loss energy due to road load, energy of vehicle inertia, and other friction and rotational inertia losses.

#### Loss Energy Due to Road Load

The road load is expressed as a quadratic formula consisting of $F_0$, $F_1$, $F_2$ terms derived through the coast-down test of the vehicle, and the road load of the test vehicle is shown in Figure 12.

The driving resistance of the vehicle is calculated using Equations (23) and (24). It was analyzed by dividing the acceleration/cruise and deceleration section for the convenience of calculation.

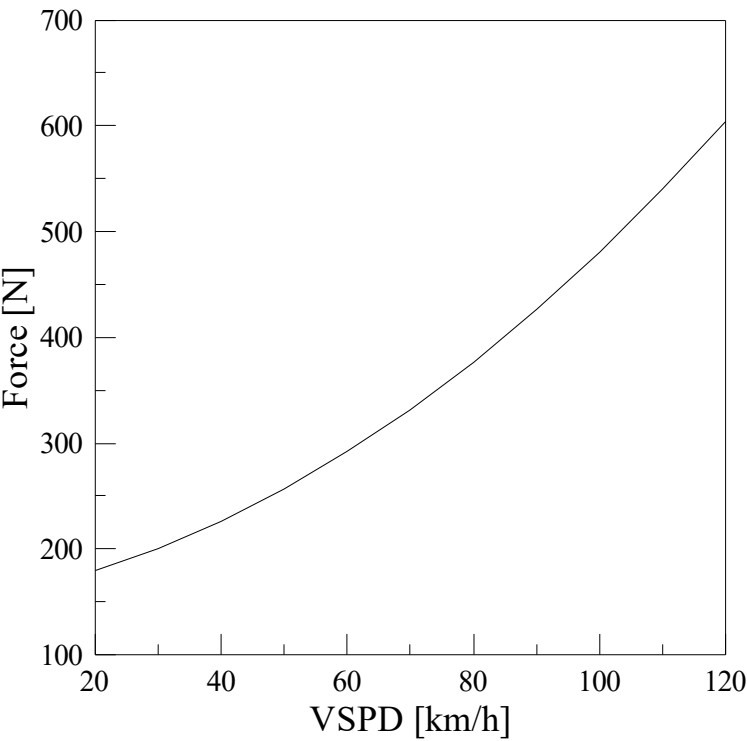

**Figure 12.** Road load of test vehicle.

Vehicle Inertia

Vehicle inertia is calculated by Equation (21), and is included as a loss term during vehicle acceleration and cruising, but during deceleration, the vehicle inertia becomes a power source to sustain vehicle operation, so the amount of lost energy is converted into regenerative braking energy as it is.

Other Losses

While the vehicle is in operation, power is transmitted from the driveshaft axle to the wheels, causing friction loss and loss due to rotational inertia. It is assumed that the rotational inertia of the wheels and axles has the greatest influence among other loss terms during vehicle acceleration and cruise. Rotational inertia was calculated as follows:

$$E_{W,I} = E_{D,out} - E_{RL} - E_{VI} \text{ (Acceleration/Cruising)} \tag{31}$$

During deceleration of the vehicle, the friction loss due to the brake pad was assumed to be the largest other loss term. Loss energy of brake pad was calculated as follows:

$$E_{Brakepad} = E_{VI} - E_{RL} - E_{D,in,Regen} \text{ (Deceleration)} \tag{32}$$

The calculated real-time energy transfer amount of each term is shown in Figure 13, and the total amount of energy at this time is shown in Table 7.

4.3.5. Energy Flow Results

Based on the results from Sections 4.3.1–4.3.4, the energy flow result of the UDDS driving cycle of an electric vehicle is shown in Figure 14, and test results are shown in Table 8.

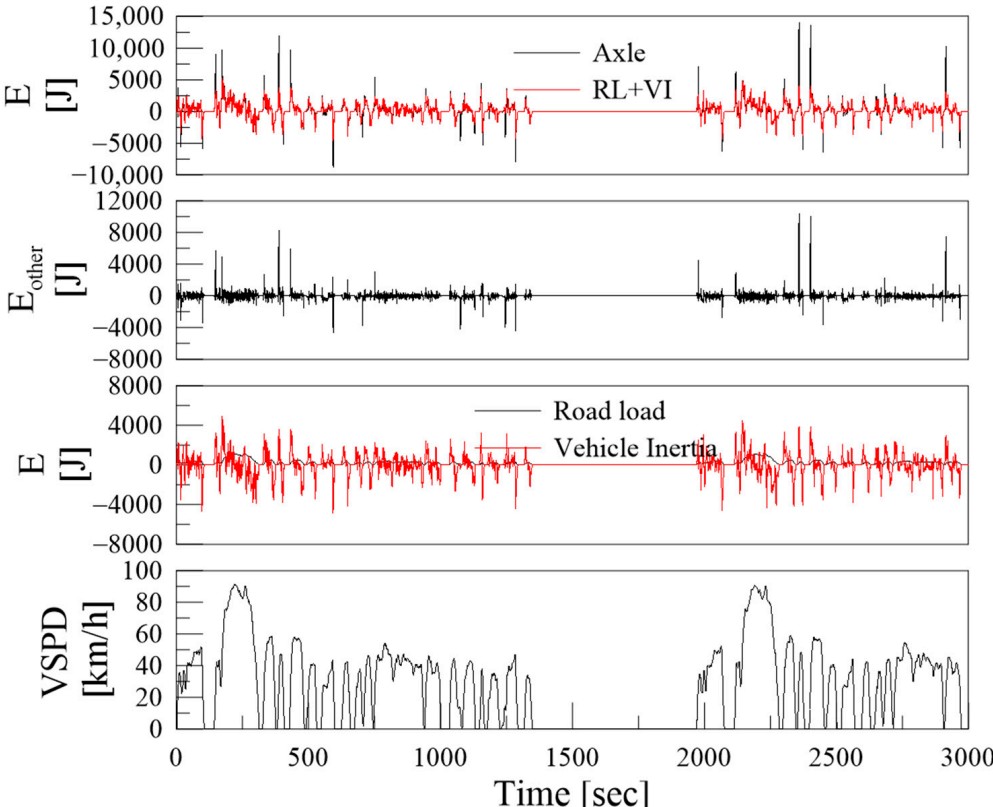

**Figure 13.** Energy transfer amounts of road load, vehicle inertia, and other losses.

**Table 7.** Total Energy amount of road load, vehicle inertia, and other losses.

| List | Energy [MJ] |
|------|-------------|
| $E_{RL}$ | 3.88 |
| $E_{RL,Regen}$ | 1.76 |
| $E_{W,I}$ | 0.07 |
| $E_{Brake\ pad}$ | 0.80 |
| $E_{VI}$ | 9.65 |

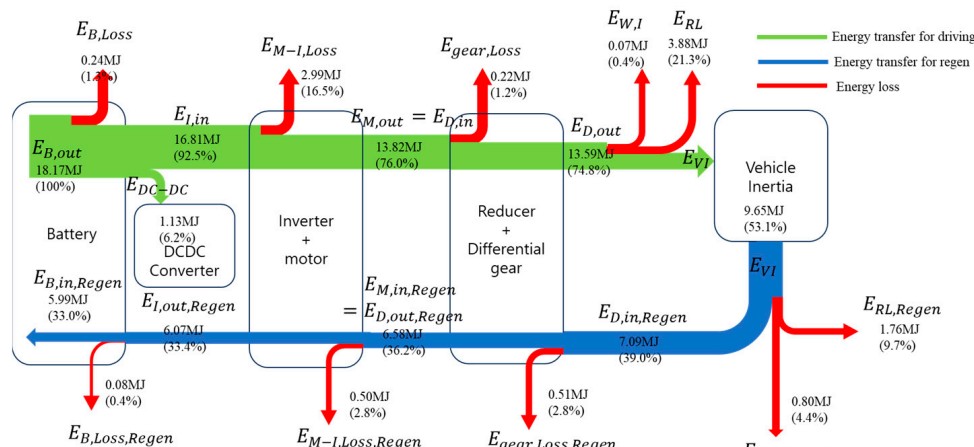

**Figure 14.** Energy flow result.

**Table 8.** Energy flow result.

| List | Value | Unit |
|---|---|---|
| Total mileage | 23.658 | km |
| $E_{Loss}$ | 12.18 | MJ |
| $E_{Battery, cap}$ | 270 | MJ |
| $SOC_{drop}$ | 4.5 | % |
| $\eta_{inverter/motor}$(Acceleration/cruising) | 82.2 | % |
| $\eta_{inverter/motor}$(Deceleration) | 92.4 | % |
| $\eta_{reducer,diff}$(Acceleration/cruising) | 98.4 | % |
| $\eta_{reducer,diff}$(Deceleration) | 92.8 | % |

Total mileage is 23.658 km, and if the mileage is calculated using the energy consumed by the battery, the energy regenerated to the battery, and the battery capacity, it is 526 km. This is a valid result, as it shows an error of 1.05% compared to the actual mileage of 530 km.

The battery–inverter electric power energy and differential–wheel power energy during vehicle acceleration, cruising, and deceleration are as follows in Table 9.

**Table 9.** Battery–inverter and differential–wheel energy in test driving cycle.

| List | Acceleration and Cruise | Deceleration |
|---|---|---|
| Battery–inverter energy [MJ] | 16.81 | 6.07 |
| Differential–wheel energy [MJ] | 13.59 | 7.09 |
| Efficiency [%] | 80.9 | 85.7 |

The efficiency in the table above can be seen as representing the integrated efficiency from the inverter to the differential. During acceleration and cruising, this efficiency is the ratio of the differential end power energy to the electric power energy of inverter. During deceleration, it is the ratio of the regenerative electric power energy of inverter to the differential end power energy. The efficiency of vehicle deceleration is higher by 4.8%, which can be attributed to the difference in efficiency of the inverter and motor between acceleration/cruising and deceleration.

Figure 9 below is a graph showing the efficiency of the inverter and motor according to the number of revolutions and the motor torque. Motor torque is a negative number during deceleration. The operation areas of the motor during acceleration/cruising and deceleration are alike, but the efficiency is high during deceleration in the same area. However, considering that the efficiency of the inverter is generally 95%, the efficiency tends to be higher than 95% when driving at reduced speed [17].

In addition, the fact that the rotational inertial energy of the motor, reducer, and differential is returned during deceleration can be seen as a factor in the difference in efficiency.

During the entire driving cycle, the electric power energy consumed by the battery, the vehicle inertia, and the regenerative electric power energy are as follows in Table 10.

**Table 10.** Battery consumption electric energy, vehicle inertia, and battery regenerative electric energy in test driving cycle.

| List | Energy [MJ] | Percentage of Battery Consumption Electric Energy [%] |
|---|---|---|
| Battery consumption electric energy | 18.17 | 100 |
| Vehicle inertia energy | 9.65 | 53.1 |
| Battery regenerative electric energy | 5.99 | 33.0 |

The total electric power energy consumed in the battery is 18.17 MJ. Of this, 53.1% is transferred to the vehicle inertial energy, and the value is 9.65 MJ. In addition, 33.0% is regenerated back to the battery through regenerative braking, and the value is 5.99 MJ.

The ratio of the electric power energy regenerated to the battery to the vehicle inertial energy is 62.1%, which is higher than the ratio of the vehicle inertial energy to the electric power energy consumed by the battery. This can be attributed to the difference in efficiency of the inverter/motor, the inertial energy of the motor and reducer/differential, and the electric power consumption of the DC–DC converter. Each energy loss item is as follows Table 11.

**Table 11.** Energy loss item analysis.

| List | Energy [MJ] | Percentage of Total Energy Loss [%] | Percentage of Total Energy [%] |
|---|---|---|---|
| Battery internal resistance | 0.32 | 2.91 | 1.74 |
| Inverter and motor | 3.40 | 31.20 | 18.70 |
| Reduction gear and differential | 0.73 | 6.73 | 4.03 |
| Road load | 5.64 | 51.80 | 31.0 |
| Friction brake | 0.80 | 7.37 | 4.41 |
| Total loss | 10.89 | 100 | 59.9 |

Energy loss has a large proportion in the order of driving resistance, inverter and motor, friction braking, reducer and differential, and battery internal resistance. The loss from road load, inverter, and motor is 83% of the total loss. Therefore, it is expected that reducing the driving resistance and the energy loss of the inverter and motor will have a great effect on reducing the total energy loss.

## 5. Discussion

In this study, the inverter and the motor were modeled together due to the difficulty of measuring data. However, if the inverter and the motor are modeled separately, it is expected that the efficiency of the inverter and the efficiency of the motor can be analyzed separately. However, since it is difficult to install a current sensor between the inverter and the motor, it is expected that it will help to more accurately analyze inverter and motor efficiency if data are obtained through preliminary tests of individual inverters and motors, and each is modeled. In addition, since the efficiency effect of the motor/inverter by the cooling flow rate of the motor and inverter and whether warm-up is applied is significant, it will be possible to obtain more accurate efficiency data if the temperature condition is additionally considered in future studies [18].

The UDDS driving test is an ideal environment test in which the acceleration and deceleration are gentle, and the outside temperature is fixed at 25 degrees Celsius. The ratio of frictional braking loss and the ratio of battery loss due to battery resistance in the UDDS test is not large compared to the total energy loss. However, if sudden braking is frequent in real road conditions, the ratio of frictional braking energy to total braking energy increases, so the frictional braking loss energy also increases [19]. Battery losses also depend on the internal resistance of battery. The internal resistance of a battery changes with temperature. In an environment where the external temperature is low or rapid acceleration is frequent, it seems that the internal resistance increases, and the energy loss increases because the temperature of the battery is too low or too high [20–22]. Therefore, friction braking and battery losses are expected to vary in magnitude depending on the driving environment. In addition, the operation of air conditioners and heaters during driving in hot and cold weather are some of the important loss factors for energy consumption of electric vehicles, but in the test drive situation of this study, the operation of the air conditioner is not considered, so the corresponding loss factors are not considered. According to previous studies, the variation in energy efficiency of electric vehicles according to temperature variations in Europe differs by about 24% [23]. This is the result of considering the average temperature of Europe, and in countries with large temperature differences, this deviation may be wider. In future research, in addition to the driving test in an ideal environment, it

is necessary to compare the effect of energy loss of each part in each environment if actual road driving and driving tests under severe conditions are conducted.

## 6. Conclusions

In this study, a model for energy flow analysis of each part of a mid-size electric passenger vehicle was presented, and based on this model, energy flow analysis was performed on an actual vehicle. In energy flow analysis, the total energy consumed by the battery is calculated through the summation of the input and output energies of the battery. The calculated energy consumption was verified by comparing the maximum mileage of the vehicle derived from the battery energy consumption pattern with the maximum mileage of the vehicle notified by the manufacturer.

During acceleration/cruising of the vehicle, the battery provides a power source, and most of the output energy of the battery is delivered to the inverter/motor. The transmitted energy is converted into mechanical power energy by the motor. The mechanical power energy is transmitted to the wheels through the reducer/differential gear and enables the vehicle to run. In this process, the energy output from the battery results in energy loss due to inverter/motor drive efficiency, friction loss in the reducer/differential gear, loss due to rotational inertia of wheel and axle shafts, and loss due to road load. About 50% of the output energy is converted into vehicle inertia energy.

During deceleration of the vehicle, the inertial energy accumulated during acceleration/cruise becomes a power source for driving the vehicle. Vehicle inertial energy can be classified into loss energy such as loss due to road load, loss due to frictional resistance of the brake pad, friction loss in the reducer/differential gear, and loss due to the generation efficiency of the motor/inverter. Finally, approximately 30% of energy discharged by the battery of vehicle is recovered through regeneration.

As a result of energy flow analysis, the loss from road load and loss of motor/inverter account for a large portion of the total loss; reducing them is expected to have a large effect on reducing the total energy loss.

The proposed energy flow model is expected to help identify and improve vulnerabilities in energy efficiency by identifying the specificity of the electric vehicle energy flow.

**Author Contributions:** Conceptualization, Y.A. and B.Y.; data curation, B.Y.; formal analysis, Y.A. and B.Y.; funding acquisition, J.P.; investigation, Y.A. and B.Y.; methodology, Y.A., J.P. and K.P.; project administration, J.L.; software, Y.A., B.Y. and K.P.; supervision, J.P., K.P. and J.L.; validation, J.P.; writing—original draft, Y.A.; writing—review and editing J.P. All authors have read and agreed to the published version of the manuscript.

**Funding:** This research was funded by Korea Ministry of Environment (MOE). (2021003390002).

**Data Availability Statement:** The data used in this study was measured as part of a technology development project supported by the Korea Ministry of Environment, and it is difficult to disclose the data with free access due to security issues in technology development. The author decides to share the data after going through a security consultation if there is a reasonable request.

**Acknowledgments:** This work was supported by the Korea Environmental Industry & Technology Institute (KEITI) through R&BD Project for management of atmosphere environment project.

**Conflicts of Interest:** The authors declare no conflict of interest.

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
