# Peer review of "Analysis of Energy Flow in a Mid-Sized Electric Passenger Vehicle in Urban Driving Conditions"

_wevj, doi:10.3390/wevj14080218_

Round 1

Reviewer 1 Report

1) Poor Quality of Figures: The paper lacks clear and well-presented figures that aid in understanding the energy flow and analysis results. The figures should be improved in terms of resolution, clarity, and labeling. The figures should also include proper axis labels, legends, and units. Clear and informative figures are crucial for readers to grasp the concepts and findings presented in the paper.

2) Insufficient Methodological Details: The methodology section of the paper lacks sufficient details regarding the modeling and calculation processes. The authors should provide a step-by-step explanation of the energy flow modeling process, including the equations and assumptions used. Additionally, the specific methods and algorithms employed in MATLAB/SIMULINK for real-time energy flow calculations need to be described in detail. Providing a comprehensive methodology will enhance the reproducibility and validity of the results.

3) Lack of Statistical Analysis: The results presented in the conclusion lack statistical analysis and confidence intervals. The authors should include statistical measures such as standard deviation or error bars to quantify the uncertainty in the calculated energy flows and losses. This will provide a better understanding of the reliability and significance of the findings. Additionally, conducting hypothesis testing or statistical comparisons, where applicable, would further strengthen the analysis.

4) Inadequate Discussion of Limitations: The conclusion briefly mentions the limitations of the study, such as the need for more reliable data and separate modeling of the inverter and motor. However, these limitations should be discussed in more detail. Provide a thorough discussion of the potential sources of error or uncertainty in the measurements, modeling assumptions, and calculation methods. Addressing these limitations will help in assessing the reliability and generalizability of the results. Additionally, suggest future research directions that could overcome these limitations and further improve the understanding of energy flow in electric vehicles.

Based on these deficiencies in the figures, results, and methodology, I recommend rejecting the paper in its current form.

Author Response

  • Poor Quality of Figures:The paper lacks clear and well-presented figures that aid in understanding the energy flow and analysis results. The figures should be improved in terms of resolution, clarity, and labeling. The figures should also include proper axis labels, legends, and units. Clear and informative figures are crucial for readers to grasp the concepts and findings presented in the paper.

       >   Fixed poor resolution pictures. (figure 2)

Label all graphs with legends and units. (figure 1, figure 14)

Added more graphs to help understand the energy flow analysis results.

(figure 3, 4, 5, 6, 7, 8, 9, 11, 12, 13)

  • Insufficient Methodological Details:The methodology section of the paper lacks sufficient details regarding the modeling and calculation processes. The authors should provide a step-by-step explanation of the energy flow modeling process, including the equations and assumptions used. Additionally, the specific methods and algorithms employed in MATLAB/SIMULINK for real-time energy flow calculations need to be described in detail. Providing a comprehensive methodology will enhance the reproducibility and validity of the results.

>    it is described in the introduction to Section 4.

3) Lack of Statistical Analysis: The results presented in the conclusion lack statistical analysis and confidence intervals. The authors should include statistical measures such as standard deviation or error bars to quantify the uncertainty in the calculated energy flows and losses. This will provide a better understanding of the reliability and significance of the findings. Additionally, conducting hypothesis testing or statistical comparisons, where applicable, would further strengthen the analysis.

> Since most of the measured signals in this study do not have data corresponding to true values, it is difficult to show statistical data such as the error bar graph you mentioned. I described the current sensor validation process in Section 4.2.2 and the torque sensor calibration data in Section 4.2.3 to ensure data reliability.
Verification of total energy loss is described in Section 4.3.5.

4) Inadequate Discussion of Limitations: The conclusion briefly mentions the limitations of the study, such as the need for more reliable data and separate modeling of the inverter and motor. However, these limitations should be discussed in more detail. Provide a thorough discussion of the potential sources of error or uncertainty in the measurements, modeling assumptions, and calculation methods. Addressing these limitations will help in assessing the reliability and generalizability of the results. Additionally, suggest future research directions that could overcome these limitations and further improve the understanding of energy flow in electric vehicles.

      > In section 5 discussion
I amended to clearly state the reasons for the limitations of the motor/inverter model and suggest method to improve them.
Suggests the possibility of deviation between mode testing in a limited environment and driving on real roads, and suggests future research directions

Reviewer 2 Report

1.                  What is the main question addressed by the research?

During an urban driving cycle, the energy flow on a mid-sized passenger electric car is examined. The study's goal is to look at how energy is generated, distributed, and used within the vehicle system under normal urban driving conditions.

2.                  Do you consider the topic original or relevant in the field? Does it address a specific gap in the field?

The study investigates the energy dynamics of mid-sized passenger electric cars in real-world urban driving settings. It fills a specific gap by focusing on energy flow analyses during urban driving cycles for this particular vehicle category. This study adds to a better knowledge of energy use and optimization in mid-sized passenger electric vehicles when driving in cities, paving the way for future advances in the sector. The topic is not wholly unique, but it fills a specific gap and expands on current knowledge in the field.

3.                  What does it add to the subject area compared with other published material?

This work adds value to existing published research by giving a complete examination of energy flow in mid-sized passenger electric cars during urban driving cycles. It gives extensive results on how energy is distributed and utilized across various vehicle components and subsystems. The study provides insights into the efficiency of energy management systems, identifies possible areas for development, and helps to optimize energy use in urban driving situations. It adds to the discipline by investigating energy flow in this unique setting, providing significant knowledge for future study and development.

4.                  What specific improvements should the authors consider regarding the methodology? What further controls should be considered?

In terms of methods, the authors might consider giving greater material on the experimental setup, apparatus, and data collecting protocols. This would improve the research's transparency and replicability. Furthermore, to offer a better understanding of the process, a full explanation of the analytical methods used, such as simulation models or measurement procedures, should be included.

5.                  Are the conclusions consistent with the evidence and arguments presented and do they address the main question posed?

The conclusions could properly clarify the significant findings about energy flow in mid-sized passenger electric cars during urban driving cycles and address the primary topic presented. They should give insights on energy distribution, use trends, and potential implications for optimizing energy efficiency. To guarantee consistency, the conclusions should be consistent with the facts and arguments given throughout the work.

6.                  Are the references appropriate?

The references are relevant and credible, and they back up the research's assertions and conclusions but they are not enough. They must be increased and new one have to  include resources on issues such as electric vehicle energy flow analysis, urban driving cycles, and energy management systems, among others.

7.          Please include any additional comments on the tables and figures.

The tables and figures in the study support the arguments and conclusions well but the quality have to be improved

Based on my analysis, I did not find any major grammatical or spelling errors in the article.

Author Response

sorry. I don't understand what exactly you point out that items 1-3 and 5 are insufficient.

Please clarify again and I will edit the manuscript.

  1. What specific improvements should the authors consider regarding the methodology? What further controls should be considered?

> Added data collection method, validation of collected data, and sensor calibration.

  1. Are the references appropriate?

> References related to recent research trends in electric vehicles have been added (4-9)

  1. Please include any additional comments on the tables and figures.

    > Improved low quality pictures and added pictures to explain the model.

Reviewer 3 Report

The topic of the article is current, I have no serious comments in terms of factual content. The structure of the computational model for the calculation of energy flows in a battery vehicle is appropriately designed, the relevant equations are given. It is not entirely clear whether some of the numerical values shown in e.g. Figures 3 and 4 were obtained only by calculation or by a technical experiment on the mentioned vehicle. How were the waveforms in Figure 6 obtained?
The article also requires a number of modifications in terms of its formal treatment.
The quality of Figure 2 is completely unsatisfactory.
References in the text should always be part of the relevant sentence, I recommend editing on lines 29, 31, 222 and others.
Entering individual items in References is confused, apparently the template was misunderstood, entries need to be modified. 

The text of the article should be generally edited by someone with an appropriate knowledge of English.

Authors incorrectly use the Saxon genitive for inanimate nouns, e.g.:
Line 60: The vehicle’s energy flow...
Line 91: The battery’s energy flow ...

There are a number of minor inaccuracies in the text, such as:
Line 166: .... rear wheels is shown Figure 3 and Figure 4.
Line 211: .... efficiency o the inverter and motor ....

Author Response

Comments and Suggestions for Authors

The topic of the article is current, I have no serious comments in terms of factual content. The structure of the computational model for the calculation of energy flows in a battery vehicle is appropriately designed, the relevant equations are given.

It is not entirely clear whether some of the numerical values shown in e.g. Figures 3 and 4 were obtained only by calculation or by a technical experiment on the mentioned vehicle.
-> Figure 3 and Figure 4 were figures related to the input/output energy of the front/rear wheel motors/inverters, respectively, but the description structure in Chapter 4 was changed to more easily explain the method of energy flow analysis.

How were the waveforms in Figure 6 obtained?

-> The method of obtaining the waveform of Figure 6 (Figure 10 after modification) has been added to section 3.1.2, and the corresponding formula is also quoted when explaining the figure.

The article also requires a number of modifications in terms of its formal treatment. The quality of Figure 2 is completely unsatisfactory.
-> The resolution of Figure 2 has been corrected.

 References in the text should always be part of the relevant sentence, I recommend editing on lines 29, 31, 222 and others.
-> The reference number description in the text has been corrected.

Entering individual items in References is confused, apparently the template was misunderstood, entries need to be modified.

-> Incorrect entries in individual references have been corrected.

Round 2

Reviewer 1 Report

No more comments. Paper can be accepted in the present form.

Author Response

No comments, thank you for your attention.

Reviewer 2 Report

The manuscript has shown improvement in addressing the remarks, to further enhance the manuscript's quality, some additional work is still needed:

- the conclusion section of the manuscript needs to be revised to achieve better coherence and clarity. It is important to avoid using numbers for different parts of the conclusion, as this can lead to confusion and make it harder for readers to follow the main takeaways.

Instead, the conclusions should be presented in a more integrated and cohesive manner, summarizing the significant findings related to energy flow in mid-sized passenger electric cars during urban driving cycles. The focus should be on providing meaningful insights into energy distribution, usage trends, and potential implications for optimizing energy efficiency in a concise and straightforward way.

- further improvements are needed for the figures in the manuscript to enhance their quality and readability. For example, Figure 2 requires additional attention to ensure it effectively conveys the information to the readers.

- the references in the manuscript should be updated to include more recent and up-to-date articles. While the current references may provide valuable insights, incorporating newer research will strengthen the credibility and relevance of the study's findings.

Based on my analysis, I did not find any major grammatical or spelling errors in the article.

Author Response

  1. In agreement with your opinion, the concluding part deletes the numerical data, presents how the energy flow in the study was analyzed, and suggests how the study will be used in the energy flow analysis of electric vehicles in the future.
  2. By enlarging the size of some small pictures to improve readability,  In the case of Figure 2, the boundary dividing the intervals has been clarified to make it easier for readers to understand.
  3. research trends and results in No. 8, 14, and 23 were added to support the assumptions used in the study. In addition, references 9-10 were added to present trends in energy efficiency analysis of electric vehicles.

Reviewer 3 Report

Dear Authors,

the article was edited according to recommendations, I have no further important comments. I wish you success in your scientific work in this interesting field.

Author Response

(The authors gave the same response as above.)
